# The Effect of Holder Pasteurization and Different Variants on Breast Milk Antioxidants

**DOI:** 10.3390/antiox12101857

**Published:** 2023-10-13

**Authors:** Réka Anna Vass, Éva Mikó, Csenge Gál, Tamás Kőszegi, Csaba I. Vass, Szilvia Bokor, Dénes Molnár, Simone Funke, Kálmán Kovács, József Bódis, Tibor Ertl

**Affiliations:** 1Department of Obstetrics and Gynecology, Medical School University of Pécs, 7624 Pécs, Hungary; 2National Laboratory on Human Reproduction, University of Pécs, 7624 Pécs, Hungary; 3Obstetrics and Gynecology, Magyar Imre Hospital, 8400 Ajka, Hungary; 4Department of Microbiology, Medical School University of Pécs, 7624 Pécs, Hungary; 5Department of Laboratory Medicine, Medical School University of Pécs, 7624 Pécs, Hungary; 6Department of Pediatrics, Medical School University of Pécs, 7624 Pécs, Hungary; 7HUN-REN–PTE Human Reproduction Research Group, 7624 Pécs, Hungary

**Keywords:** glutathione, TAC, ORAC, transferrin, calcium, total protein, breast milk, donor milk

## Abstract

Background: After birth, breast milk (BM) is a known essential source of antioxidants for infants. We analyzed the non-enzymatic total antioxidant capacity (TAC), oxygen radical absorbance capacity (ORAC), and glutathione, calcium, transferrin, and total protein levels of human breast milk before and after Holder pasteurization (HoP). Methods: The collected donor BM samples were pasteurized with HoP. Results: HoP decreased TAC (−12.6%), ORAC (−12.1%), transferrin (−98.3%), and total protein (−21.4%) levels; HoP did not influence the glutathione concentration, and it increased the total calcium (+25.5%) concentration. Mothers who gave birth via Cesarean section had significantly lower TAC in their BM. TAC and glutathione levels were elevated in the BM of mothers over the age of 30. BM produced in the summer had higher glutathione and calcium levels compared to BM produced in the winter. The glutathione concentration in term milk samples was significantly higher in the first two months of lactation compared to the period between the third and sixth months. The transferrin level of BM for female infants was significantly higher than the BM for boys, and mothers with a BMI above 30 had increased transferrin in their samples. Conclusions: Antioxidant levels in human milk are influenced by numerous factors. Environmental and maternal factors, the postpartum age at breast milk collection, and Holder pasteurization of the milk influence the antioxidant intake of the infant.

## 1. Introduction

Human breast milk (BM) provides all the necessary macro- and micronutrients and many of the non-nutritive bioactive molecules needed for an infant’s development, survival, and well-being [1]. Antioxidants are compounds that help to protect the body against damage caused by free radicals, which are unstable molecules that harm cells and contribute to the development of disease. Reactive oxygen species (ROS) contribute to biological homeostasis and play a significant role in cell signaling in both psychological and pathophysiological processes, but they also cause molecular or even cell damage, necrosis, apoptosis, and DNA oxidation [2,3]. Increased oxidative stress is a known causative factor for mortality, bronchopulmonary dysplasia (BPD), and retinopathy of prematurity (ROP), especially in very-low-birth-weight infants [4,5,6]. Total antioxidant capacity (TAC) refers to the overall antioxidant capacity of a substance, including the combined effects of various antioxidants. In the context of breast milk, non-enzymatic TAC represents the collective antioxidant capacity of the antioxidants present in breast milk [7,8]. These antioxidants include vitamins (such as vitamin C, vitamin E, and beta-carotene), minerals (such as selenium and zinc), and other bioactive compounds (such as bilirubin) [9]. The non-enzymatic antioxidants in breast milk protect the infant’s cells from oxidative damage, support the developing immune system, and support physiological development [6]. The oxygen radical absorbance capacity (ORAC) assay measures a fluorescent signal that is quenched in the presence of ROS [10]. BM TAC can be estimated with ORAC, a standardized and validated method to measure the antioxidant capacity in biological samples in vitro. It assesses the ability of antioxidants to neutralize free radicals and reduce oxidative stress [8,9,10]. BM contains a dynamic and diverse array of anti- and prooxidants, and their interactions affect the ORAC level [8].

Glutathione is a powerful antioxidant that is naturally present in the human body. It plays a crucial role in protecting cells from oxidative damage caused by free radicals and toxins. Glutathione plays a crucial role in supporting the immune system by regulating immune responses, thereby enhancing the function of immune cells. In breastfed infants, glutathione in breast milk may contribute to their ability to handle and eliminate certain toxins more effectively [11,12].

Transferrin is a protein that plays a vital role in iron transport and absorption [13]. While transferrin is primarily found in the blood, it is also present in small amounts in breast milk [14]. Transferrin in breast milk helps facilitate the absorption and utilization of iron by the infant. It is necessary for the production of red blood cells and for the proper functioning of various enzymes and metabolic processes. Once absorbed, iron is utilized for various physiological processes, including the production of hemoglobin and the support of overall growth and development. Iron is an integral component of many proteins and enzymes most relevant in our context of peroxidase and catalase [15]. Iron concentrations in human milk are low (0.2–0.4 mg/L), yet the iron in milk is highly bioavailable [16].

The calcium concentration in breast milk does not change with the stage of lactation. A previous review and metanalysis reported no significant differences in milk calcium concentration between lactation stages, adolescent and adult mothers, preterm and term infants, exclusive and mixed breastfeeding, with or without calcium supplementation, between nutritional statuses, country income categories, continents, and measurement methods in calcium concentration [17].

Breast milk is important for premature and mature infants as it provides the nutrients and immune-boosting factors needed for their development and helps to reduce the risk of complications [18]. Antioxidants in BM directly influence the development of the gastrointestinal tract and, through absorption, impact organ maturation and infant development [2,6,7]. Since their intrauterine development is disrupted due to preterm birth, preterm infants are predisposed to oxidative stress, and they need certain maternal protective factors. After birth, BM is the exclusive source of these maternal protective compounds. When own mother’s milk is not available, donor milk is considered the best feeding alternative.

## 2. Materials and Methods

In this study, registered and approved donor mothers of the Breast Milk Collection Center (BMCC) (Unified Health Institution at Pécs, Hungary) were recruited, who, following the center’s protocol, donated freshly pumped milk (n = 122). Our study was conducted with the approval of the Regional and Local Research Ethics Committee of the University of Pécs, Pécs, Hungary (PTE KK 7072-2018). Waivers for participant consent were obtained. For analysis, 3 mL was poured and stored separately at −80 °C until laboratory measurements. The protocol of the BMCC was followed during our study. Our aim was to examine the effect of HoP on donor milk samples. Pool sizes were variable from 4 to 11 samples; we examined 14 pools. Figure 1. shows the experimental design (Figure 1.)

The samples were analyzed first individually and after they were pooled and Holder pasteurized (30 min at 62.5 °C) in the laboratory of the Unified Health Institution. We took five samples for later analyses; three samples were used in the present experiment. All samples from the pooled and Holder pasteurized donor milk were stored at −80 °C until laboratory measurements were taken. First, we sonicated the BM samples and centrifuged them at 15,000× *g* for 15 min. The skimmed milk was transferred for analysis according to the previously described preparation methods [19,20]. For the measurement of glutathione, every sample was analyzed, while the other factors were detected in the first 6 pools.

For TAC determination, two different assays were used: enhanced chemiluminescence (ECL) and ORAC. In the ECL assay, a fully validated luminol-peroxidase-4-iodophenol-hydrogen peroxide-based technique was applied [10]. In all analyses, first 20 μL of blank/standard/sample and then 270 μL of horseradish peroxidase-ECL reagent were pipetted into 96-well white optical plates (Optiplates, Per-Form Hungaria Ltd., Budapest, Hungary). Then, 20 μL of hydrogen peroxide solution was injected into the wells by a Biotek Synergy HT plate reader (Agilent, Santa Clara, CA, USA) and the developing luminescence signals were monitored kinetically for 10 min. Measurements were carried out in duplicate. A standard curve for Trolox calibrators was established by using the area under the curve (AUC) of the luminescence signals, and the TAC of the samples was calculated from the equation of the standard curve. The TAC values of the samples were given as Trolox equivalent in μmol/L or mmol/L.

For the ORAC technique, 25 µL of blank/standard/sample was mixed with 150 µL Na_2_- fluorescein in a black optical plate (Optiplates), and 25 µL of AAPH oxidant (2,2′-azo-bis (2-amidinopropane) dihydrochloride, Merck, Darmstadt, Germany) was injected into the wells by the Biotek Synergy HT plate reader. Kinetic measurement of fluorescence quenching was performed at 490/520 nm wavelengths for 80 min [10]. For the calculation, the AUC values obtained for the blanks/standards/samples were used, and the TAC was calculated as described for the ECL assay. The BM samples were diluted 20-fold with tri-distilled water, and measurements were carried out in duplicate. The TAC, ORAC, calcium, transferrin, and total protein concentration were detected in 59 breast milk samples at the fully accredited Department of Laboratory Medicine, University of Pécs.

The total glutathione level was detected with a colorimetric detection kit (ThermoFisher Scientific, Frederick, MD, USA) based on the manufacturer’s instructions. Next, 50 μL standards or samples were added to the wells. After additional steps, the absorbance was read at 405 nm, and the concentrations were expressed in mM. The total glutathione level was measured in 122 samples.

Calcium, transferrin, and total protein levels were measured using a fully automatized Cobas c analyzer system (Roche Diagnostics, Mannheim, Germany). The lower limit of detection for calcium was 0.20 mmol/L, for transferrin, it was 1.5 mg/L, and for total protein, it was 40 mg/L.

To test the data normality, Shapiro–Wilk tests were performed with GraphPad (La Jolla, CA, USA). Paired *t*-tests or *t*-tests were used for further analysis. The repeated measures one-way ANOVA test with the post hoc Dunnett’s test was applied to compare the effect of Holder pasteurization. Differences were considered statistically significant if the *p*-values were <0.05. The study was powered to detect moderate effect sizes (Cohen’s d = 0.6). The results are presented as the mean ± SEM. Maternal age and body mass index (BMI), infant gender, seasonal differences, and duration of lactation at the time of sampling were also analyzed.

## 3. Results

For the measurement of TAC, ORAC, transferrin, calcium, and total protein, we analyzed 59 BM samples. The mean maternal age was 32.2 ± 0.6 years, the mean BMI was 25.2 ± 0.5, and the mean infant gestational age was 38.3 ± 0.4 weeks. In total, 25 BM samples were donated by mothers who had undergone a Cesarean section (CS), and 34 samples were collected after spontaneous delivery. Female infants were delivered by 27 mothers, and 32 mothers gave birth to male infants.

The total glutathione level was measured in 122 BM samples. The average maternal age of the donors for these samples was 32.8 ± 0.4 years, the average maternal BMI was 26.4 ± 0.6, and the average gestational age of the newborns was 38.9 ± 0.2 weeks. In this study, there were 52 samples donated after CS, while after spontaneous delivery, 70 samples were donated. Out of 122 samples, 57 were produced for female infants and 65 were produced for males.

After HoP, their glutathione levels did not change; however, their TAC, ORAC, total protein, and transferrin levels were significantly lower after HoP. Their calcium concentration was higher after HoP (Table 1).

The transferrin concentration was higher in the BM produced for female infants than in the BM produced for male infants. The glutathione, TAC, ORAC, calcium, and protein levels were similar between the two groups. Mothers delivering spontaneously had significantly higher TAC concentrations than mothers giving birth by CS; otherwise, the mode of delivery did not influence the other antioxidants in breast milk. Mothers whose BMI was above 30 after delivery and during breastfeeding had higher BM transferrin content than mothers with a BMI under 30. Maternal BMI had no impact on glutathione, TAC, ORAC, calcium, or protein levels. Maternal age did not influence ORAC, calcium, protein, and transferrin content, but the total glutathione and TAC concentrations were significantly higher in the BM of mothers above the age of 30 (Table 2).

The breast milk samples collected during the winter (n = 28) had significantly lower glutathione levels than the samples collected during the summer (n = 31). The calcium concentration was significantly lower in samples collected during the summer (winter n = 17; summer n = 21) (Figure 2).

In the term breast milk samples, the glutathione concentration was significantly higher during the first two months (n = 51) of lactation compared to the period between the third and sixth months (n = 71) of breastfeeding (Figure 3).

## 4. Discussion

Breastfeeding is a critical aspect of postnatal adaptation; it plays a crucial role in providing optimal nutrition and promoting bonding and emotional attachment between the mother and child. Breast milk is highly nutritious and provides numerous benefits for the infant’s growth and development. It contains a balance of essential nutrients, antibodies, enzymes, and other bioactive compounds that support the immune system, digestive health, and overall well-being. Breastfeeding is associated with a lower incidence of a variety of oxidative stress-related illnesses in premature infants [21]. Also, it is known to be a rich source of glutathione, which contributes to the antioxidant protection provided to infants during breastfeeding. Numerous studies have proven the differences between own mother’s milk, donor milk, and infant formula [22,23]. All of them reported poorer bioactive factor, hormone, and immunoglobulin content in formula compared to donor milk or own mother’s milk. Poorer growth and developmental outcomes have been reported in infants receiving pasteurized donor milk compared to infants receiving unpasteurized human milk [24,25]. The value of antioxidant richness in BM is conceivably important to protect nursing infants against oxidative stress [26,27].

Preterm infants are exposed to a wide range of stressors, e.g., blood tests, infections, phototherapy, oxygen supplementation, parenteral nutrition, and therapeutic interventions during their care. Although ROS contribute to homeostasis, they also participate in cell signaling both in physiological and pathophysiological processes, and ROS can cause molecular and cell damage [2,3]. An investigation found that the phototherapy of jaundiced neonates resulted in increased oxidative stress [28]. As an explanation for the decrease in antioxidant capacity of BM with HoP, thermally induced denaturation is the most likely mechanism. HoP is known to change the composition of BM and decrease the concentration of different hormones and compounds [1,19,20,22]. Other preservation techniques, like refrigeration, also change the composition of BM [20]. A previous study showed that the antioxidant activity of BM decreased significantly from the 21st day of cold storage (at 4 °C and −20 °C) [29]. A recent meta-analysis of the effect of HoP on the antioxidant properties of human milk showed inconclusive results regarding the effect of pasteurization on the TAC of BM [7]. Some studies proved a reduction in TAC after HoP compared with untreated BM, while others detected no influence of Holder pasteurization [8].

Glutathione is a known antioxidant in human milk. It deactivates oxygen-derived free radicals and eliminates toxins, carcinogens, and malonic dialdehyde [30]. Silvestre et al. investigated the effects of HoP, and in contrast to our results, they found that HoP reduced glutathione concentrations in human milk by 46%. They investigated the effect of high-temperature, short-time pasteurization and described no concentration changes in glutathione levels after the procedure [31]. The antioxidant levels of colostrum, mature milk, and transitional milk are different [32]. Our results showed that the glutathione content of BM decreases with time. The ability of the neonatal intestine to mitigate radical accumulation plays a role in its capacity to overcome oxidative stress. Lipid peroxidation is known to preferentially target polyunsaturated fatty acids, and oxidative injury of necrotizing enterocolitis leads to deregulation of the glutathione defense system [11].

BM’s non-enzymatic TAC is believed to play a significant role in protecting the infant from oxidative stress and supporting its overall health and development. TAC is a general measure to indicate the level of free radicals scavenged by a test solution that is commonly used to assess the antioxidant status of human milk samples. TAC provides a general assessment of the antioxidant capacity of a given bioactive component, while the quantification of nutrient antioxidants, specific antioxidant enzymes, conjugated dienes, or lipoprotein oxidation provides more specific information [33,34,35]. With age, BM melatonin concentration was shown to be decreased [36]. No previous data were found about TAC or total glutathione levels in BM related to maternal age. Our present results suggest a correlation between maternal age and TAC and glutathione concentration, which might be a compensatory mechanism of age-related processes. However, it is important to understand that the precise impact and significance of non-enzymatic TAC in BM on infant health are still areas of ongoing research.

In a previous study, the BM of mothers with gestational diabetes had similar ORAC levels compared to non-diabetic mothers. BM ORAC was positively correlated with BM ascorbic acid in mothers with gestational diabetes [8]. Colostrum showed significantly higher ORAC values compared with mature milk [32].

The calcium concentrations detected in our study were similar to previous results [37,38]. The HoP calcium concentration was found to be elevated; presumably, the heat treatment unbound the calcium in the BM. This phenomenon was observed with Il-7 [39] and TSH [20] in previous studies. No differences in calcium concentration were detected based on gestational age [40]. A review found that conditions like familial hypophosphatemia and hyperparathyroidism affect BM calcium concentrations, but other environmental parameters did not influence calcium concentration [41]. Minerals, like calcium in milk, are particularly important for infant skeletal development and may reflect maternal characteristics [42]. In a previous study on children, compared to winter, children in the spring and summer had significantly lower plasma calcium concentrations [43]. The absorption of calcium is controlled by vitamin D from the small intestine [44]. Theoretically, the serum calcium concentration in the summer is higher than in the winter; we did not find dietary differences among the food consumption of the involved women. Seasonal diversity in T cell activity may also be associated with seasonal changes in blood calcium levels [45]. These results suggest that other factors may influence the calcium content of BM. The total protein level of BM was not affected by HoP in our present study, in agreement with previous reports [23,46,47].

A limited amount of information is available on the presence of transferrin in breast milk [48,49]. The present work shows that HoP vigorously decreases the concentration of transferrin in BM. BM produced for female infants contains higher levels of transferrin than milk produced for boys. An earlier study reported that infant girls had higher hemoglobin and serum ferritin concentrations than boys [50]. The serum ferritin level was found to be elevated in individuals with increased BMI values [51], while in our study, the transferrin level was higher in the BM of mothers with a BMI above 30.

It is important to note that the content of antioxidants and other components in human milk may vary depending on various factors such as the mother’s diet, overall health, stage of lactation, and mode of delivery. Additionally, the pasteurization process, such as HoP, is known to affect the levels of certain components in human milk. However, despite any potential changes, human milk remains an excellent source of nutrition and immune protection for infants [52,53]. Earlier findings shed light on how the hormonal components of milk have sex-specific effects on offspring growth during early postnatal life with varying temporal windows of sensitivity [54,55]. The total dietary antioxidant capacity of patients’ diets significantly depended on the season and was highest in the summer [56]. It is known that maternal diet influences the composition of BM [51,52,57,58] and may result in epigenetic changes [59]. Our results demonstrate that the antioxidant content of BM is influenced by the seasons and may reflect maternal diet as well.

The antioxidant properties of human milk limit the consequences of excessive oxidative damage. After birth, especially in premature infants, the gastrointestinal tract is under development, which leads to incomplete or slow protein digestion [60], promoting the absorption and bioavailability of BM components. Previous research has demonstrated that the addition of antioxidants to infant formula increases infant resistance to oxidative stress [27,61]. Continuous ROS exposure can induce metabolic changes such as hyperglycemia in extremely-low-birth-weight infants [62].

Hormone levels (e.g., leptin) [63] and supplementation (e.g., thyroxin) [64] show a connection with developmental outcomes in early childhood, suggesting that continuous monitoring of antioxidant levels or supplementation during intensive care should be investigated in clinical trials. Anti- and prooxidants control a sensitive balance in newborns via multiple factors, such as immunoglobulins, short-chain fatty acids, and cytokines (which can be found in BM), and, through absorption and local effects, influence their reaction to intensive care treatment, survival, and development [65,66,67,68]. The impact of oxidative stress and the balance or imbalance of pro- and antioxidants control intrauterine development and postnatal life [69,70,71]. Obstetrical complications, like the premature rupture of membranes [72] or preeclampsia [73], also intensify OS-induced processes. During postnatal adaptation oxidative stress may have an impact on adult diseases affecting the cardiovascular or endocrine system [74,75]. OS, similar to inflammation, promotes aging-related pathologies, endothelial dysfunction, and adverse pregnancy outcomes [76]. Maternal nutrition influences the nutritional programming of the offspring; in later life, cardiovascular diseases, metabolic syndrome, diabetes, insulin resistance, glucose intolerance, fertility issues [77,78], and hypertension may develop in adulthood [79]. Dysfunction of hypothalamic appetite control results in obesity through increased lipogenesis [80]. Preterm infants with a higher total antioxidant status are more likely to be protected from free radicals, blocking ROS accumulation and OS [81]. BM is a known source of antioxidant capacity, providing and supporting breastfed preterm neonates [82]. Antioxidant treatment has become a potential and predictably essential therapeutic strategy in the treatment of preterm newborns with bronchopulmonary dysplasia [83,84] or necrotizing enterocolitis [85,86]. The prevention of chronic morbidities of extremely premature newborns by different therapeutic options and adjuvant perinatal strategies are already highlighted in clinical practice [87,88,89]; therefore, supplementing antioxidants during intensive care should be investigated.

Our study has limitations. We only examined the chosen antioxidants in BM samples before and after HoP but revealed higher transferrin levels in the BM produced for female infants and in the BM of mothers with BMIs of 30 or above. Our results suggest that some antioxidants are present in higher concentrations in BM in the summer than in the winter. Elevated maternal age was associated with higher glutathione and TAC levels in BM, and after vaginal delivery, glutathione was present in higher concentrations in BM than after Cesarean section. Our results have augmented our knowledge about the effects of HoP: glutathione concentration was not impacted by HoP, but TAC, ORAC, and transferrin levels were reduced.

## 5. Conclusions

Breast milk is considered the most optimal feeding option for an infant, with it having greater benefits in the case of preterm birth [90,91,92,93]. Breastfeeding influences the rate of OS bias postnatal adaptation, through impacting insulin sensitivity, choline and prostaglandin metabolism, and lipid profile during early infancy [94,95,96]. Although HoP modifies the antioxidant content of BM, donor milk is still considered the most suitable alternative to a mother’s own BM. HoP reduced the TAC, ORAC, and transferrin concentration in BM. The clinical significance of the changes in BM antioxidants with HoP is unknown; further research is necessary to improve our knowledge. Transferrin levels are higher in BM produced for female infants, the TAC concentration was found to be elevated in BM after vaginal delivery, and an elevated maternal BMI resulted in higher transferrin levels. Our present results demonstrate that the antioxidant intake of the infant is influenced by the gender of the infant and maternal and environmental factors. Knowing that these antioxidant compounds actively influence physiological processes, antioxidant supplementation and guidance of human milk banks in testing different pasteurization processes in order to maximize the preservation of antioxidant properties are highly recommended.

## Figures and Tables

**Figure 1 antioxidants-12-01857-f001:**
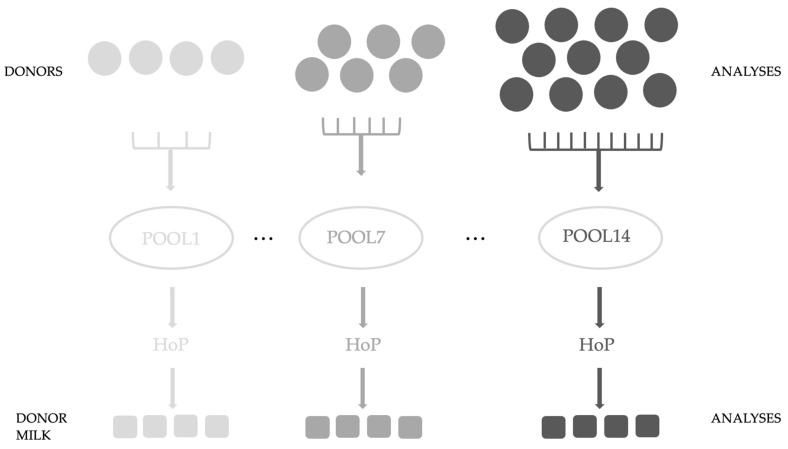
Experimental design of our study.

**Figure 2 antioxidants-12-01857-f002:**
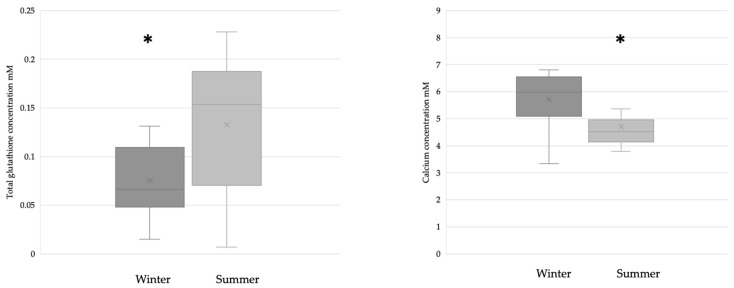
Glutathione (winter n = 28; summer n = 31) and calcium (winter n = 17; summer n = 21) levels in BM samples collected during the winter and summer. In the case of glutathione, * *p* = 0.0167; in the case of calcium, * *p* = 0.0128.

**Figure 3 antioxidants-12-01857-f003:**
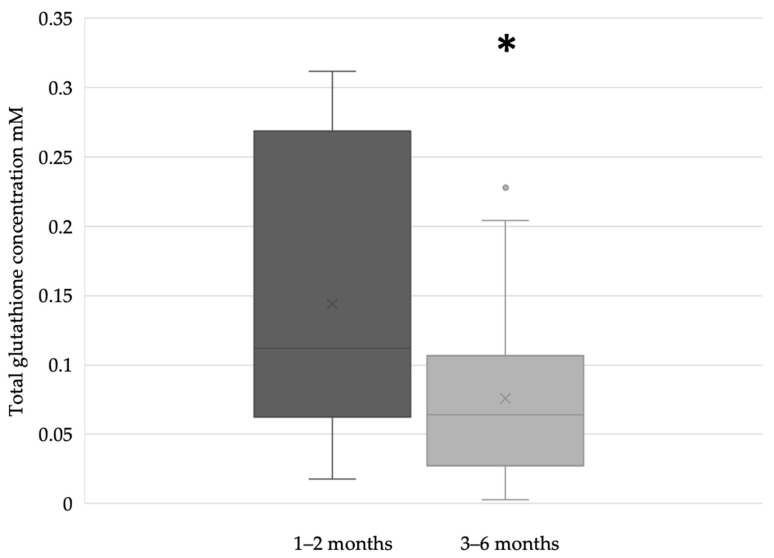
Glutathione concentration of term breast milk during the first 6 months of lactation (* *p* = 0.0213).

**Table 1 antioxidants-12-01857-t001:** The effect of Holder pasteurization (HoP) on different factors.

	Raw	HoP	*p*-Value
Glutathione (mM) (n = 112)	0.11 ± 0.02	0.12 ± 0.10	0.575
TAC-ECL (μM) (n = 59)	127.31 ± 6.24	111.27 ± 5.26	0.028
TAC-ORAC (μM) (n = 59)	3602.34 ± 104.13	3163.13 ± 787.94	0.001
Calcium (mM) (n = 59))	5.08 ± 0.15	6.38 ± 0.10	<0.0001
Total protein (g/L) (n = 59)	4.91 ± 0.16	3.86 ± 0.11	<0.0001
Transferrin (mg/L) (n = 59)	80.82 ± 10.85	1.34 ± 0.09	<0.0001

HoP: Holder pasteurization; TAC: total antioxidant capacity; ORAC: oxygen radical absorbance capacity.

**Table 2 antioxidants-12-01857-t002:** Antioxidants and total protein in human milk.

		Glutathione mM	TAC µM	ORAC µM	Calcium mM	Total Protein g/L	Transferrin mg/L
Infant gender	Girl	0.08 ± 0.01 (n = 57)	134.06 ± 8.08 (n = 27)	3752.71 ± 122.34 (n = 27)	5.21 ± 0.21 (n = 27)	5.02 ± 0.22 (n = 27)	**108.39 ± 16.62** (n = 27)
	Boy	0.15 ± 0.05 (n = 65)	114.82 ± 9.83 (n = 32)	3406.66 ± 192.41 (n = 32)	4.78 ± 0.22 (n = 32)	4.74 ± 0.27 (n = 32)	**44.22 ± 8.54 *** (n = 32)
Delivery	Vaginal delivery	0.13 ± 0.04 (n = 70)	**142.07 ± 7.96** (n = 25)	3733.45 ± 149.79 (n = 25)	5.05 ± 0.25 (n = 25)	5.07 ± 0.25 (n = 25)	69.1 ± 16.5 (n = 25)
	C-section	0.08 ± 0.01 (n = 52)	**109.94 ± 8.75 *** (n = 34)	3507.41 ± 153.97 (n = 34)	5.02 ± 0.17 (n = 34)	4.77 ± 0.24 (n = 34)	94.16 ± 15.68 (n = 34)
Maternal BMI	<30	0.12 ± 0.03 (n = 90)	130.21 ± 7.07 (n = 41)	3623.63 ± 132.36 (n = 41)	4.99 ± 0.21 (n = 41)	4.84 ± 0.23 (n = 41)	**52.8 ± 7.94** (n = 41)
	>30	0.07 ± 0.01 (n = 32)	105.97 ± 13.83 (n = 18)	3605.48 ± 192.65 (n = 18)	5.12 ± 0.24 (n = 18)	5.06 ± 0.24 (n = 18)	**139.87 ± 24.30 *** (n = 18)
Maternal age	<30	**0.06 ± 0.01** (n = 51)	**99.61 ± 9.36** (n = 34)	3690.39 ± 193.75 (n = 34)	5.15 ± 0.28 (n = 34)	4.65 ± 0.25 (n = 34)	81.63 ± 20.04 (n = 34)
	>30	**0.11 ± 0.01 *** (n = 71)	**134.07 ± 7.56 *** (n = 25)	3592.23 ± 134.22 (n = 25)	4.99 ± 0.19 (n = 25)	5.01 ± 0.22 (n = 25)	82.76 ± 13.93 (n = 25)

***** *p* < 0.05. TAC: total antioxidant capacity; ORAC: oxygen radical absorbance capacity; BMI: body mass index; C-section: Cesarean section.

## Data Availability

Data applied in this study are available from the corresponding author upon request.

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
