# Peer review of "The Effect of Holder Pasteurization and Different Variants on Breast Milk Antioxidants"

_antioxidants, 2023, doi:10.3390/antiox12101857_

Round 1

Reviewer 1 Report

"Effect of Holder pasteurization on breast milk antioxidants" by Réka Anna Vass and co-workers directed to Antioxidant/MDPI is focused on the changes observed in antioxidant levels in HoP-treated human milk. The quality of breast milk and changes in its composition is an important problem. In this manuscript, the authors presented different research groups of breastfeeding women to show what factors may influence the variation in breast milk quality, mainly antioxidant.

As a reviewer, I have some general and detailed comments on the manuscript.

I would ask the authors to complete the information on the authorisation of the collection of milk from mothers. Please state the authorisation number obtained from Ethical Commission and write what the inclusion and/or exclusion criteria were taken into consideration for the milk donors in the study. How much milk was collected for the study from one woman? Did the women were informed and signed the participation agreement to be involved in the study?

Tab.2 – Please, change the title of the table, as it shows not only antioxidants but also other components.

Fig 1,2 - Please describe the x-axis.

Line 140-148 –The authors indicated that they had taken milk from different numbers of mothers for different analyses:

“For the measurement of TAC, ORAC, transferrin, calcium, and total protein we analyzed 59 BM samples”. “Total glutathione level was measured in 122 BM samples.”

However, in the article, the results of these analyses are placed in one table (Tab. 1 2), which may distort the assessment of the actual state of affairs. I would ask you to clarify this situation and make a correction.

Line 171- would you be so kind and try to explain, why is so significant difference in calcium concentration in winter and summer milk? What is the reason for such a large difference?

Summing up, after correction the article will be suitable for publication in Antioxidants/MDPI.

Author Response

Dear Reviewer 1,

We thank your thorough work, we have strengthened our manuscript based on your suggestions, please find our responses below.

I would ask the authors to complete the information on the authorisation of the collection of milk from mothers. Please state the authorisation number obtained from Ethical Commission and write what the inclusion and/or exclusion criteria were taken into consideration for the milk donors in the study. How much milk was collected for the study from one woman? Did the women were informed and signed the participation agreement to be involved in the study?

Thank you, we added information between Line 92-96. The Breast Milk Collection Center has a protocol based on international guidelines, including examinations and instructions for donor mothers during the donation period. We did not apply further exclusion or inclusion criteria since our aim was to examine the antioxidant content of breast milk in everyday clinical practice, since this BMCC supplies the Neonatal Intensive Care Unit of the University of Pécs. Mothers pumped the donated BM into sterile bags, and the transport of BMCC collected the samples at a scheduled time. Mothers fulfilled a questionnaire and received a handout about the project, and we contacted the mothers on the phone or personally and after they signed the application form accepted by the Regional and Local Research Committee of the University of Pécs (PTE KK 7072-2018).

Tab.2 – Please, change the title of the table, as it shows not only antioxidants but also other components.

Thank you, we corrected the title.

Fig 1,2 - Please describe the x-axis.

Thank you, the figures have been changed.

Line 140-148 –The authors indicated that they had taken milk from different numbers of mothers for different analyses:

“For the measurement of TAC, ORAC, transferrin, calcium, and total protein we analyzed 59 BM samples”. “Total glutathione level was measured in 122 BM samples.”

However, in the article, the results of these analyses are placed in one table (Tab. 1 2), which may distort the assessment of the actual state of affairs. I would ask you to clarify this situation and make a correction.

We added the exact N value to each component in the table.

Line 171- would you be so kind and try to explain, why is so significant difference in calcium concentration in winter and summer milk? What is the reason for such a large difference?

We did not find any connection between maternal diet and calcium intake. Further information was added to the text between Lines 266-272.

Summing up, after correction the article will be suitable for publication in Antioxidants/MDPI.

Thank you.

Sincerely,

Reka Vass MD PhD

Reviewer 2 Report

The authors have presented their research work with a proposal based on studying the effect of Holder pasteurization on breast Milk oxidants. Results describe other statistical differences related with age, diet, season, months of lactation, etc. HoP is a minimal part of this study, and authors need to revise title and/or objective of the paper in order to clarify how to describe and discuss their results. It could be published after major revisions. Additional information need to be included (see comments):

·       BMI (body mass index) is not defined in the text.

·       Section 2. Material and Methods. Author’s text needs to be clarified. “(n=122). Samples were analyzed individually, after they were pooled, and Holder pasteurized”

How many samples were included in each pool? How many pools were prepared? Please clarify the reasons to prepare those pools with specific samples.

·       Results section. According text (lines 140-148) Glutatione was the only one parameter measured in all samples. Other parameters were analysed in 59 samples. Please clarify this data in material and methods section including a scheme, figure or table and additional explanation. Include other relevant information such as BMI values.

·       Table 1 (indicate samples in each study). Glutatione n=122, other parameters n=59

·       Table 2. Change title table to clarify content. Include in table the number of samples included for each parameter or type of sample.

Significant digits need to be revised.

·       Discussion. In this section authors evaluate their results in comparison with other authors. They include data such as effect of diet, health, age, … However, there is scarce information to justify data exposed in Results section when when comparing data before and after HoP.

·       Conclusions only include general comments, and they need to be rewritten focusing the most significate findings of this study.

Other comments:

English should be revised

Use concentration abrevations properly (line 73      mg/L)

English needs to be improved

Author Response

Dear Reviewer 2,

We thank the careful revision of our work; we have strengthened our manuscript based on your suggestions.

The authors have presented their research work with a proposal based on studying the effect of Holder pasteurization on breast Milk oxidants. Results describe other statistical differences related with age, diet, season, months of lactation, etc. HoP is a minimal part of this study, and authors need to revise title and/or objective of the paper in order to clarify how to describe and discuss their results. It could be published after major revisions. Additional information need to be included (see comments):

  • BMI (body mass index) is not defined in the text.

BMI is first mentioned in line 140.

  • Section 2. Material and Methods. Author’s text needs to be clarified. “(n=122). Samples were analyzed individually, after they were pooled, and Holder pasteurized”

How many samples were included in each pool? How many pools were prepared? Please clarify the reasons to prepare those pools with specific samples.

The protocol of the Breast Milk Collection Center (BMCC) was followed during our study. The present work followed the everyday practice of BMCC, our main aim was to examine the effect of Holder pasteurization on donor milk samples. Pool sizes were variable from 4 to 11 samples, we examined 14 pools. We added new information between Lines 91-95.

  • Results section. According text (lines 140-148) Glutatione was the only one parameter measured in all samples. Other parameters were analysed in 59 samples. Please clarify this data in material and methods section including a scheme, figure or table and additional explanation. Include other relevant information such as BMI values.

The be more transparent, the exact sample number was added in the tables.

  • Table 1 (indicate samples in each study). Glutatione n=122, other parameters n=59

Thank you, we added this information to Table 1.

  • Table 2. Change title table to clarify content. Include in table the number of samples included for each parameter or type of sample.

Thank you, we’ve changed the title and included the N values.

Significant digits need to be revised.

We revised the tables.

  • Discussion. In this section authors evaluate their results in comparison with other authors. They include data such as effect of diet, health, age, … However, there is scarce information to justify data exposed in Results section when when comparing data before and after HoP.

Discussion was amended focusing on these suggestions.

  • Conclusions only include general comments, and they need to be rewritten focusing the most significate findings of this study.

This part of the text was rearranged.

Other comments:

English should be revised

Prof. Dr. Edward F. Bell revised the manuscript.

Use concentration abrevations properly (line 73      mg/L)

Thank you, we have corrected it.

Sincerely,

Reka Vass MD PhD

Round 2

Reviewer 2 Report

The manuscript has been improved. However, there are some questions needing additional revisions:

1.      The number of pools has not been included in the reviewed version. Data marked in red has not been included in the reviewed version.

“The protocol of the Breast Milk Collection Center (BMCC) was followed during our study. The present work followed the everyday practice of BMCC, our main aim was to examine the effect of Holder pasteurization on donor milk samples. Pool sizes were variable from 4 to 11 samples, we examined 14 pools. We added new information between Lines 91-95.”

2.      In section 2 line 98 authors confirm that Samples were analyzed after, they were pooled,. In the previous response (1) authors confirm that they have examined 14 pools. However, in the manuscript, (line 152) and in Table 2, authors describe data from more that 14 samples (pools). Please clarify pools and single samples and studies involved with one and other (pools and individual samples). Please explain in Material and Methods section your experimental design including samples and polls involved, and all the assays carried out with each sample/pool.

Author Response

Dear Reviewer 2,

we thank your thorough work , please find our responses below.

  1. The number of pools has not been included in the reviewed version. Data marked in red has not been included in the reviewed version.

“The protocol of the Breast Milk Collection Center (BMCC) was followed during our study. The present work followed the everyday practice of BMCC, our main aim was to examine the effect of Holder pasteurization on donor milk samples. Pool sizes were variable from 4 to 11 samples, we examined 14 pools. We added new information between Lines 91-95.”

Answer: This part was added to the manuscript in the Materials and methods section.

  1. In section 2 line 98 authors confirm that Samples were analyzed after, they were pooled,. In the previous response (1) authors confirm that they have examined 14 pools. However, in the manuscript, (line 152) and in Table 2, authors describe data from more that 14 samples (pools). Please clarify pools and single samples and studies involved with one and other (pools and individual samples). Please explain in Material and Methods section your experimental design including samples and polls involved, and all the assays carried out with each sample/pool.

Answer: Figure 1. presents our experimental design to clarify the details, and also other parts were added to inform the readers about the details.

Thank you for revising our manuscript.

Sincerely,

Reka Vass MD PhD